# Machine Learning for Plant Stress Modeling: A Perspective towards Hormesis Management

**DOI:** 10.3390/plants11070970

**Published:** 2022-04-02

**Authors:** Amanda Kim Rico-Chávez, Jesus Alejandro Franco, Arturo Alfonso Fernandez-Jaramillo, Luis Miguel Contreras-Medina, Ramón Gerardo Guevara-González, Quetzalcoatl Hernandez-Escobedo

**Affiliations:** 1Unidad de Ingeniería en Biosistemas, Facultad de Ingeniería Campus Amazcala, Universidad Autónoma de Querétaro, Carretera Chichimequillas, s/n km 1, El Marqués CP 76265, Mexico; amanda.rico@uaq.mx (A.K.R.-C.); miguel.contreras@uaq.mx (L.M.C.-M.); 2Escuela Nacional de Estudios Superiores Unidad Juriquilla, UNAM, Querétaro CP 76230, Mexico; alejandro.francop@unam.mx; 3Unidad Académica de Ingeniería Biomédica, Universidad Politécnica de Sinaloa, Carretera Municipal Libre Mazatlán Higueras km 3, Col. Genaro Estrada, Mazatlán CP 82199, Mexico; afernandez@upsin.edu.mx

**Keywords:** eustress, crop improvement, intelligent algorithms, agricultural engineering

## Abstract

Plant stress is one of the most significant factors affecting plant fitness and, consequently, food production. However, plant stress may also be profitable since it behaves hormetically; at low doses, it stimulates positive traits in crops, such as the synthesis of specialized metabolites and additional stress tolerance. The controlled exposure of crops to low doses of stressors is therefore called hormesis management, and it is a promising method to increase crop productivity and quality. Nevertheless, hormesis management has severe limitations derived from the complexity of plant physiological responses to stress. Many technological advances assist plant stress science in overcoming such limitations, which results in extensive datasets originating from the multiple layers of the plant defensive response. For that reason, artificial intelligence tools, particularly Machine Learning (ML) and Deep Learning (DL), have become crucial for processing and interpreting data to accurately model plant stress responses such as genomic variation, gene and protein expression, and metabolite biosynthesis. In this review, we discuss the most recent ML and DL applications in plant stress science, focusing on their potential for improving the development of hormesis management protocols.

## 1. Introduction

Stress is a defensive state of an organism resulting from deviations of its optimal developmental conditions [1]. Environmental challenges destabilize fundamental biological functions in plants, and this is often perceived as constrained crop growth and development. In agricultural systems, the detrimental effects of plant stress are a significant cause of productivity loss, threatening food security, especially in the current context of climate change [2]. Nevertheless, aside from being deleterious, stress responses can also induce desirable traits in crops and therefore be considered favorable [3]. In that case, stress is often called eustress, a term derived from the Greek prefix *eu* that means good or well [4].

Whether a stressor will harm or benefit an organism depends entirely on the intensity of its incidence [5]. This observation derives from the fact that plant defensive responses to stress are biphasic, meaning that high doses of a stressor tend to be unfavorable, whereas low doses are beneficial [6] (Figure 1). The phenomenon that explains such biphasic behavior is called hormesis, and it demonstrates how some level of stress is necessary for a plant to achieve optimal fitness [7].

The acquaintance of hormetic responses of plants to stressors is the basis for implementing hormesis management, which refers to the deliberate exposure of crops to eustress for eliciting desirable attributes [8]. This practice is also called controlled elicitation, and it can considerably increase crop yield, growth, quality, pest resistance, and overall stress tolerance [9,10]. Nevertheless, the design of hormesis management protocols still faces many limitations, mainly due to the complexity of plant stress responses and the lack of consideration of the hormesis model in plant stress research [7].

Several factors shape the defensive responses of plants, including the type of stressor, the genetic identity of the individual, its developmental stage, its nutritional stage, and the responding tissue or cell [11]. Consequently, any given dose of a specific stressor will fall within a different dose-response range according to the responding organism and the observed output variable [4]. Therefore, eustress doses must be experimentally determined before proposing their implementation [12,13]. Moreover, until very recently, studies on plant stress responses usually focused on determining a damaging-dose threshold rather than depicting the whole dose-response curve [6,7,13]. For these reasons, applying novel technologies for enhancing the scopes of hormesis management in agriculture is crucial [14].

Artificial intelligence (AI) is an evolving branch of computer science with great potential to solve a variety of complex problems of the modern world. From using advanced fuzzy logic models for wastewater treatment [15], estimating the production of biosurfactants by bacteria with artificial neural networks (ANN) and the fuzzy inference system ANFIS [16], to advanced Deep Learning tools in plant science. In that matter, AI tools are helpful to model plant distribution, identify species, recognize disease and stress, diagnose nutritional deficiencies, and apply agrochemicals in precision agriculture [17]. In particular, Machine Learning (ML) techniques can predict the outcome of various complex biological processes, such as gene function, gene networks, protein interactions, and optimal growing conditions, leading to significant achievements in plant stress research [18,19].

Literature reviews assessing the use of machine learning for plant stress research focus mainly on analyzing the numerous findings on the identification, classification, quantification, and early prediction of deleterious plant stress responses [20,21,22,23]. However, the potential of the most recent modeling techniques for understanding eustress responses for crop improvement remains unexplored. Therefore, this review aims to present the most recent findings in plant stress research and propose machine learning to model dose-response for implementing hormesis management in agriculture.

## 2. Hormesis and Plant Stress Science

The hormesis model is rapidly gaining recognition and acceptance within academia for depicting the responses to external stimuli of any given biological system [24]. Nevertheless, the concept of hormesis is not recent. The first reports to describe the biphasic dose-response date from the late 19th century, whereas the term hormesis appeared in the literature for the first time in 1943 [14]. Despite its early description, the scientific community rejected the hormesis phenomenon until the last decades of the 20th century because it was mistakenly associated with homeopathy [25]. Consequently, most of the research around dose-response falls under the assumption of linearity of such responses and overlooks low-dose stimulation, which impacts how scientists, regulatory agencies, and society conceive stress [6]. However, hormesis appears in stress studies at such a high frequency that it has quickly regained consideration in the design of research projects [26].

At present, low-dose stimulation has multiple applications in clinical medicine, environmental risk assessment, ecology, crop management, and sustainable agriculture [27,28]. Several findings suggest that hormetic responses are highly generalizable and occur in all kinds of biological systems [29,30]. Moreover, the quantitative characteristics of hormesis remain constant among models as they correspond to the limits of biological plasticity, meaning that hormesis is related to adaptability and evolution [31]. This latter fact is especially relevant for understanding plants, given that their survival relies entirely on their adaptive responses to stressors due to their sessile nature.

Plants possess complex defensive mechanisms to deal with the many biotic and abiotic challenges they may encounter in the wild. Protected agriculture diminishes these challenges causing plants to underdevelop defensive mechanisms and making them more susceptible to environmental stress and pests or decreasing their production of desirable specialized metabolites [32]. In this scenario, deliberately exposing crops to low-dose stress may enhance plant productivity and stress resistance [3]. However, controlled stimulation of plant defensive mechanisms leads to many different observable outcomes as plants can sense different stressors and respond to them in a specific manner [33].

The specificity of defensive responses depends directly on the plant species, the type of stressor, and the responding tissue. As a result, plant stress responses are diverse, and so are the methods for their analysis [34]. Each stress response is a multilayered molecular process that can be understood as an information transfer between the stressor perception and the expression of a phenotypic trait. Calabrese and Blain [30] assessed more than 3000 hormetic plant responses with numerous response variables, each representing a specific point in the physiological pathway triggered by the stress incidence. Such observations show that, given their complexity, analyzing plant physiological mechanisms can generate a significant amount of data.

## 3. Data in Plant Hormesis Research

The analysis of a substantial number of pathosystems has permitted the description of many receptor-to-response routes [35]. However, to picture plant immunity as a collection of independent downstream cascades is now thought to be quite simplistic [36]. In contrast, plant immunity should rather be understood as an intricate systemic molecular network capable of simultaneously perceiving biotic and abiotic stressors and ultimately leading to transcription reprogramming and protective physiological responses [37,38,39].

The convenience of choosing one variable or another as the output of hormesis management depends on the target crop, the cultivation conditions, and the productivity objective. For example, increasing the drought tolerance of a food crop cultivated in a controlled environment would be irrelevant, whereas augmenting its yield would be paramount. Therefore, the description of plant defensive responses must consider several endpoints (response variables) to possess technological significance. Fortunately, collecting a considerable amount of data from biological systems is becoming more frequent, thanks to the latest advances in analytical instrumentation and techniques [40].

High-throughput analyses increase the chances to elucidate physiological processes and ecological interactions of plants from the broadened perspective of systems biology [41]. The generation of big data sets from the simultaneous analysis of an extensive collection of biomolecules corresponding to a definite category (genes, transcripts, proteins, and metabolites) has led to the so-called omics approach, which is the primary tool of systems biology [42]. Furthermore, a multi-omics approach makes it possible to obtain a more detailed snapshot of a plant system by simultaneously analyzing its whole genome, proteome, transcriptome, and metabolome [40]. Moreover, the multi-omics approach applied to single-cell functional analyses can simplify data processing and modeling to accurately depict many biological processes in plants [43].

In the following subsections, we will briefly describe the main types of data each omics approach can deliver when applied to plant hormesis research.

### 3.1. Genomics and Transcriptomics

Genomics refers to the sequencing, assembly, and functional analysis of the genome of a plant, and it has advanced more rapidly than any other omics in plant science [44]. Only in the last two decades, the sequences of more than 100 plant genomes have been published, and further technological advances in genomics have increased our understanding of plant biology leading to substantial agricultural progress [45].

Genome sequencing has several applications in plant stress science. The structural analysis of DNA is not only fundamental for classifying organisms but also for identifying stress-driven mutations, which occur in plants under heat [46], drought [47], and other abiotic stresses [48]. Moreover, DNA structural variations occurring under low-dose stress can be linked to gene function using gene ontology analyses to reveal the genetic basis of hormesis [49]. DNA sequence variations such as Single Nucleotide Polymorphisms (SNPs) are also helpful for understanding the molecular mechanisms underlying hormetic responses when analyzed along with phenotypic traits as in Genome Wide Association Studies (GWAS) [50,51]. GWAS analyses consider big data sets to identify and predict gene candidates and quantitative trait loci accountable for stress responses [52].

SNPs genotyping in combination with other data sets from high-throughput analyses such as phenomics or enviromics has also led to the development of genomic selection for optimizing crop breeding [53]. With this strategy, it is possible to improve physiological traits with hormetic behavior in crops, such as yield, pest resistance, and environmental stress tolerance, to shorten breeding cycles and decrease the need for continuous phenotyping [54]. Additionally, the advent of outstanding new genome-editing techniques, such as the Zinc Finger Nucleases (ZFN), the Transcription Activator-Like Effector Nucleases (TALENs), and the Clustered Regulatory Interspaced Short Palindromic Repeats (CRISPR) systems, implies, along with transcriptomics, the most significant advance in the development of stress-resistant crops [55].

The rapid advances in sequencing technologies and bioinformatics have also substantially impelled RNA analyses [56]. The synthesis of RNA is dynamic, depending on the activation of a gene to occur. Therefore, transcriptomics is the key to investigating gene function in targeted physiological mechanisms qualitatively and quantitatively [57]. Detecting hormetic stimulation at the transcript level can be achieved by analyzing the differential expression of known genes on small (~20) [58,59,60] and very large scales using microarray technology (~50,000) [61] or by completely sequencing the RNA from a sample as in next-generation and third-generation sequencing [62,63,64].

The computational analysis of transcriptomic datasets precedes the reconstruction of gene regulatory networks and the crosstalk by which they interconnect during specific physiological processes [65]. In particular, machine learning algorithms can infer interactions between genes with great accuracy [66]. Nevertheless, gene regulation during hormetic responses involves biological processes other than transcription, such as epigenome dynamics, which depends on chromatin structural changes, namely DNA methylation and histone modifications [67]. Therefore, integrating additional types of datasets and adding spatial and temporal information is fundamental to increasing model resolution and depicting the mechanisms of hormesis truthfully [68].

### 3.2. Proteomics

Proteins are the main regulatory molecules in every cell process. Therefore, the ensemble of differentially translated proteins in response to a given stimulus is an additional dataset that contains essential information for ascertaining hormetic cellular mechanisms [69]. Moreover, many studies show that RNA quantity does not proportionally relate to protein abundance [70]. The latter occurs mainly due to additional regulation steps between transcription and protein synthesis and the stability of the end products [71]. For that reason, transcriptomics and proteomics, or other high-throughput analyses should be simultaneously conducted to validate and reconstruct entire regulatory networks [72].

Detecting differential changes in plant proteome is especially useful for studying plant stress responses since relatively small variations in the dose of a stressor result in a significant difference in the proteome at both mild and severe stressor incidence [73]. Furthermore, under stress conditions, the plant cell upregulates the expression of proteins associated with primary metabolic processes such as photosynthesis, redox homeostasis, energy metabolism, nitrogen absorption, and the biosynthesis of signaling molecules [74,75]. Therefore, plant proteomics can help researchers detect stress at a molecular level earlier than it would be possible by analyzing changes in observable phenotypic traits and for both stress-susceptible and tolerant genotypes [76].

Proteome analyses make it possible to identify and characterize novel proteins, and along with bioinformatics, proteomics enables tracking variations in protein abundance, form, cellular location, and activity following a stressors incidence [77]. Additionally, proteome research has proven helpful for clarifying cellular organelle function, post-translational modifications, and protein–protein interactions, providing a more in-depth insight into the stress-driven molecular mechanisms of plant cells [78]. Proteomics technologies range from the classic gel-based and the Liquid Chromatography coupled to Mass Spectrometry (LC-MS) approaches to the modern Mass Spectrometry Imaging (MSI), and combined with additional high-throughput analyses, these still underexploited tools are among the most powerful methods for unraveling the molecular mechanism of hormetic stress responses in plants [79].

### 3.3. Metabolomics

A number of the differentially expressed proteins resulting from stress incidence are regulators that activate and shape specialized metabolic pathways inside the cell [80]. Metabolomics is the study of all the small molecules in a tissue, which, in the case of plants, possess a unique structural and functional complexity [81]. Moreover, due to their sessility, plants depend on chemical signaling to maintain homeostasis and ecological interactions at intra- and interspecies levels. Plant specialized metabolism is evolutionarily shaped by environmental pressures to synthesize chemical compounds with an enormous structural and functional diversity and capable of interacting with living tissues [82].

Many plant specialized metabolites are an active part of plant internal signaling pathways or exert bioactivity on other organisms [83]. Furthermore, every plant organism is a genuine biological factory capable of synthesizing an estimated 30,000 metabolites [84]. Hence, plants are a significant source of chemical compounds with pharmacological properties and are particularly valuable among natural products for drug discovery purposes [85]. In addition, plant metabolites are fundamental for maintaining human health by conferring nutritional, functional, and nutraceutical value to food products [86].

Being an adaptive response, the activation of plant metabolism also exhibits a hormetic behavior [87,88,89], and deliberately exposing crops to low-dose stress is a convenient means for stimulating metabolites accumulation [90]. Moreover, the metabolomics analysis of plant stress response along with bioinformatics makes it possible to find candidate markers for directing crop breeding and predicting crop performance under environmental stress [91].

Given the structural diversity of plant metabolites, the main limitation of metabolomics resides in developing comprehensive extraction techniques and analytical methods to detect a big heterogeneous ensemble of chemical compounds simultaneously. Nevertheless, thanks to the recent advances in coupled analytical technologies and bioinformatics, particularly Mass Spectrometry (MS), Nuclear Magnetic Resonance (NMR), and hybrid MS/NMR methods [92], it is now possible to separate and detect the whole metabolome from a biological sample quickly and affordably [93]. Moreover, many intrinsic experimental conditions for metabolome analysis are compatible with other omics studies, making metabolomics a convenient foundation for designing and fulfilling multi-omics experiments and an effective tool for systems biology research [94].

### 3.4. Phenomics

Plant phenotyping is the measurement of phenotypic traits either at the cell, organ, or whole plant level for understanding the underlying mechanisms of the interactions of plants with their environment [95]. Molecular responses drive phenotypic change, and for that reason, the developmental traits of plants, such as growth, seed germination, photosynthesis, transpiration, stomatal conductance, and pigmentation, among others, also display hormetic behavior [96]. As a result, various sensors can be used for differentially analyzing physiological plant hormetic responses to stress-related events [97].

Image-based phenotyping is useful to detect leaf morphological variations in plants [98]. Red-green-blue (RGB) imaging uses Charge Coupled Device (CCD) or Complementary Metal Oxide Semiconductor (CMOS) sensors to detect color changes related to plant stress responses. Such sensors work within the visible range of the electromagnetic spectrum and are convenient to diagnose nitrogen (N), phosphorous (P), potassium (K), magnesium (Mg), calcium (Ca), and iron (Fe) deficiency symptoms [99]. Detecting nutrient deficit can also help identify environmental stress incidence. For example, Martinez et al. (2020) [100] reported that water deficit modifies nitrate uptake by altering the expression of genes related to nitrate assimilation in the roots and the shoot. Moreover, changes in pigment content can be related to visible stress symptoms in such a detailed manner, that it is possible to discriminate between their biotic or abiotic origin [101].

Yellowing is the most notable symptom of leaf senescence, and it appears due to seasonal developmental processes, pathogen attack, and abiotic stressors incidence, indicating a decrease in the photosynthetic rate [102]. Chlorophyll metabolism is regulated in a hormetic manner, and therefore it can perform as a biomarker to identify other metabolic changes resulting from low-dose stress incidence [103,104]. Many imaging techniques focus on detecting chlorophyll fluctuations with convenient results for biotic and abiotic stress phenotyping, such as chlorophyll fluorescence. This technique is relevant to determining overall crop fitness, and due to its high sensitivity, it has been extensively applied for the early detection of stress incidence [105].

Imaging techniques can also be used to analyze plant physiological processes and identify stress even in the absence of symptoms unobservable for the unaided eye. Magnetic resonance imaging can be applied to plant systems to elucidate plant-water relationships and as a post-harvest control to determine maturity and mechanical damage of agricultural products [106]. Thermography uses optical sensors that detect radiation outside the visual range of the electromagnetic spectrum, and it has been used to detect plant interactions with biotic and abiotic stressors and monitor environmental stress susceptibility and resistance [107,108]. Mild-stress responses can also be detected using multispectral and hyperspectral imaging. Multispectral imaging considers only specific bands of electromagnetic spectra, whereas hyperspectral imaging increases the resolution of the wavelengths. These technologies can identify plant diseases such as yellow rust and powdery mildew in wheat and leaf rust in sugar beet from early stages [109]. Moreover, multispectral imaging works on large scales by employing uncrewed aerial vehicles and satellites. Therefore, multispectral and hyperspectral imaging, along with other omics techniques, could be used to develop hormesis management protocols at a crop scale.

Given the intricacy of physiological responses, the elucidation of the adaptive mechanisms of plants to low-dose stress must be carried out from a multidimensional approach, utilizing comprehensive analyses for detecting the differential changes stimulated in different layers of the stress response. Understanding such mechanisms and, in particular, characterizing the hormetic dose-response curve allows eustress treatments to be implemented to enhance stress tolerance and increase food production and quality [7]. Nevertheless, integrating multiple-layer datasets gives rise to additional challenges beyond data collection and storage, including data management and processing [110]. Therefore, handling and modeling hormetic responses from multi-omics data requires computational methods for transforming data into knowledge.

## 4. Artificial Intelligence Applications in Plant Stress Science

High-throughput analyses of functional molecules such as genes, transcripts, proteins, and metabolites, produce a tremendous amount of data from biological systems. However, without proper processing, such data lack biological significance. Therefore, the advances in analytical methods and instrumentation have also generated the need for processing tools capable of describing mechanistic associations and interactions. The persisting escalation of computing power has triggered a diversification of artificial intelligence (AI) tools to address various problems in plant science. AI algorithms are remarkably advantageous to identify and classify individual characteristics within an extensive set of experimental data, and thus they are a promising means for analyzing plant stress mechanisms [104]. Furthermore, if we consider the accumulating evidence on the hormetic behavior of plant stress responses, intelligent algorithm applications in plant stress physiology could be helpful for predicting eustress responses that fall under the low-dose stimulation model.

A considerable number of AI applications on plant stress research implement Machine Learning (ML) and its subtype Deep Learning (DL). Such techniques have been applied in the paradigm of the four categories for analyzing the process of plant stress: identification, classification, quantification, and prediction (ICQP) [23]. Most of the published image-processing phenotyping studies use ML and DL tools to identify and classify stress symptoms, whereas the prediction of phenotypic traits before their expression is the most frequent application when analyzing genomic and transcriptomic datasets [111]. Figure 2 shows the four categories of the ICQP paradigm and the different applications for plant stress research integrating each category.

ML and DL techniques have extended the reach of traditional statistics for modeling non-linear systems such as biological processes [112]. Hence, both tools effectively process data to analyze plant responses to biotic and abiotic stressor incidence. Table 1 comprises recent studies examining plant stress using ML and DL techniques. Table 1 comprises some recent studies examining plant stress using ML and DL techniques. This summary was created by searching the academic databases ScienceDirect, Springer link, IEEE Xplore, and Google scholar. This review is up to date until January 2022, covering the work carried out from 2016 to 2022. The keywords used for this search were “artificial intelligence”, “machine learning”, “deep learning”, “plant stress”, “plant disease”, “plant resistance”, and “plant science”. It includes a classification of current research on plant stress elucidation using ML tools, emphasizing the algorithms used, the ICQP paradigm category on which each report lies, the stressors studied, and the datasets analyzed.

The majority of the studies listed in this literature revision are based on supervised algorithms whose quality depends on the data sources, the feature extraction from the available data, and the selection of the output variables and the learning algorithms [139]. The most frequently reported algorithms for successfully modeling plant stress responses are the Support Vector Machine (SVM), Random Forest (RF), and Convolutional Neural Networks (CNN). These techniques were predominantly used for identifying and classifying stress symptoms from extensive images datasets. Nevertheless, machine learning algorithms have also been used to predict many other various stress dose-dependent traits, such as disease-resistance gene expression [140], transcription factor expression [141], yield [142], growth [143], and specialized metabolites biosynthesis [144].

### Deep Learning Platforms and Potential Applications for Plant Hormesis Management

Given that data available for understanding biological systems continues growing, the complexity of AI systems keeps increasing, and thus new high-performance artificial neural networks (ANN) were needed for the cases in which conventional ML techniques have fallen short. Such networks are currently known as deep learning, a technology that consists of the assembly of machine learning algorithms, increasing the number of levels and non-linear transformations in the neural networks and the efficiency of the training process [145].

Before the current heyday of DL, a disinterest towards it existed for several years, mainly due to hardware limitations and lack of funding. However, these techniques were reassessed as soon as more powerful hardware became available, especially Graphics Processing Units (GPUs) [146]. These devices were initially designed to compute three-dimensional graphics in video games and proved to be good performers of parallel computing; therefore, such systems were promptly used for processing DL algorithms. Nowadays, thanks to the multicores of modern GPUs allowing more and faster operations, DL is one of the most powerful AI tools to model complex non-linear systems and process an enormous number of experimental data.

The use of high-performance software allows DL to be used to evaluate many problems in systems biology. Computational models have been applied to simulate protein interactions, a universal process in biological systems, and a key process for understanding the mechanism of physiological responses [147]. Using in silico prediction of protein–protein and protein–ligand interactions, it is possible to estimate the activity of potential effector molecules and effectively test complete libraries with millions of molecules without performing costly and time-consuming experiments [148]. For plant systems, in silico prediction could offer a relevant platform for analyzing receptor binding during stressor perception and identifying new phytohormones or designing elicitors to achieve optimal responses for hormesis management [149].

When it comes to DL, there is an impression that an expensive computational infrastructure is necessary, which is not entirely false considering the cost of high-performance GPUs. However, there are many platforms for training DL models through cloud computing. For example, platforms such as Amazon Web Service (AWS), Google Colab, Microsoft Azure provide CPUs, Hard Disks, GPUs, or TPUs through the command terminal, which allows one to utilize high-performance hardware from an average computer with internet access and sufficient bandwidth. Additionally, some of these platforms offer a free version, meaning that cloud computing can be performed for a few hours without cost. This approach avoids the cost of a local computing cluster, the specialized space to house them, and the required electricity, all considerable limitations to perform this kind of computing due to the long time needed to train a DL model at a high performance. Table 2 shows the platforms or hardware used to train DL models applied to current plant science research.

In addition to exploiting hardware features, it is also essential to take account of the existing software to ensure an adequate running performance when training ML networks. Several platforms enable the implementation of ML and DL algorithms, and many of the available frameworks are open-source software, which has led to the rapid adoption of computer modeling for many agricultural technology [160]. Among the most employed is Caffe, a deep learning framework that has been developed by Berkeley AI Research (BAIR) and community contributors. There is also high-level software developed from C++ and C code, such as Open CV, and an increasing number of programs based on Python, such as Keras, Pytorch, scikit-learn, and Theano [161]. Moreover, the Microsoft Cognitive Toolkit (CNTK), used as a library or standalone ML tool, and TensorFlow, available from Google Brain, are two of the most widespread open-source platforms for executing DL tasks. Finally, it is also indispensable to consider the tools offered by Matlab and the Nvidia CUDA software to implement AI applications in agriculture [162].

Hormesis management increases crop yield and quality, stimulates specialized metabolism, and enhances stress tolerance [8]. Nevertheless, characterizing hormetic curves for several species and evaluating multiple stressor doses to produce an expected physiological response is a slow and expensive process. In addition, controlled elicitation studies show that a considerable crop extension or high technology greenhouses are necessary to evaluate the effect of low-dose stress for developing eustress management protocols [163,164]. Furthermore, it is necessary to consider thousands of individual plant markers such as genotype information, yield performance, and environmental data to propose effective treatments.

DL would be an efficient tool to address hormesis management limitations because it can expose complex non-linear relations between environmental conditions and gene expression to decipher gene networks and signaling pathways [165]. Convolutional Neural Networks (CNN) applied to image analysis is one of the most used biometric techniques in agriculture for evaluating plant identity, morphology [157,158], growth [159], disease [153,154,156], and pollution [150]. The CNN architecture is designed as a matrix for data analysis. It is structured so that, at several stages, filters segment the data and acquire specific information to train the deep neural network [166]. Modern plant analysis techniques can easily detect significant variables related to stress mechanisms with enough sensitivity for characterizing hormetic fluctuations. CNN could be trained with such datasets to model crop performance and predict phenotypic variables in response to low-dose stress (Figure 3).

## 5. Limitations, Challenges, and Future Outlook

In this review, we have discussed the advantages of ML for assessing research problems in plant-stressor systems from a subcellular to an ecosystem scale. Nevertheless, there are also relevant limitations to consider for proposing the implementation of ML tools, particularly for hormesis modeling.

Firstly, given the vast diversity of ML and DL platforms, selecting an appropriate architecture to carry out the proposed strategy constitutes a significant challenge. Furthermore, every architecture performs differently depending on the number and type of deep networks and the running hardware, complicating, even more, the tool selection. In response to this challenge, the number of scientific publications discussing ML tool-pairs, their performance, and new models designed specifically to perform a given task constantly increases [167].

The second challenge to assess refers to AI’s fundamental limitations. The power of ML methods offers advantages over conventional statistics, but they do not explicitly infer or provide confidence boundaries such as p values. This is a problem since scientists commonly rely on confidence intervals and model interpretation to support decision-making. Moreover, increasing the complexity of the network architecture turns ML systems into “black-boxes.” Consequently, most ML and DL methods do not allow a straightforward interpretation or the basis for the resulting predictions [168]. Therefore, an evaluation method after a training process is crucial. A high network intricacy and cumulative input datasets, central to analyzing plant systems, also require more computing power, significantly increasing the time and cost to complete tasks. The latter mainly affects multifaceted algorithms such as SVM and MLP [169,170].

The third challenge arises from the input data. The success of ML depends on the availability of appropriate databases, that is, extensive collections of data sharing specific features [111]. Nonetheless, although hormesis is increasingly being considered in plant research protocols, it still lacks the attention needed to form public databases to enrich model training. Moreover, data from biological systems are highly heterogeneous, and, as a result, detailed data curation and preprocess must be performed to ensure the accuracy of the training process [171]. Furthermore, even if multilayer data sets from plant hormesis research were available, there are simply too many different plant species interacting with changing environments. As a result, any group of experimentally acquired data results partial and unrelated to others. The use of model species is fundamental to depict basic biological processes, but these findings are not always transposable to evolutionarily distant plants or other species of interest. For these reasons, plant scientists must agree and standardize research methods for describing hormetic responses at all levels in representative plant species far and wide the phylogenetic tree. Considering the state-of-the-art discussion and the challenges that arise from conceiving the present proposal, the flow process depicted in Figure 4 conceptualizes the application of ML to model hormetic responses of plants to controlled stress exposure. However, there are still some constraints regarding the lack of experimentally adequate data to develop a robust model that could facilitate eustress doses determination and ultimately optimize hormesis management implementation for improving crop performance. For these reasons, future work on plant stress should emphasize the hormesis model and the construction of public knowledge databases, including plant phenotyping results and validated tools, models, and platforms.

## 6. Conclusions

The aim of elucidating plant stress responses is to develop cost-effective methods for producers to manipulate plant systems and obtain desirable phenotypes. Nevertheless, given the diversity of the technologies and methods currently available to measure variables associated with plant stress responses, the standardization of the experimental conditions and the integration of different dataset collections is a significant challenge. Moreover, most of the research around stress focuses on the adverse effects it causes on the plant system and completely ignores eustress and the hormetic behavior of plant defense.

Interestingly, it could be possible to develop robust models of plant responses if we consider that the behavior of stress responses is generalizable and varies within the limits of biological plasticity rather than depending only on the genetic identity or the developmental stage of individual systems [172]. However, even if we assess physiological responses from a hormetic approach, the big data challenge remains. In this respect, hormesis research should capitalize on the strengths of ML and DL for developing models capable of utilizing experimental data to predict which actions are required to improve crops traits. The latter would be especially beneficial when the eustress dose ranges of a stressor are unknown, and datasets from related crops are available.

## Figures and Tables

**Figure 1 plants-11-00970-f001:**
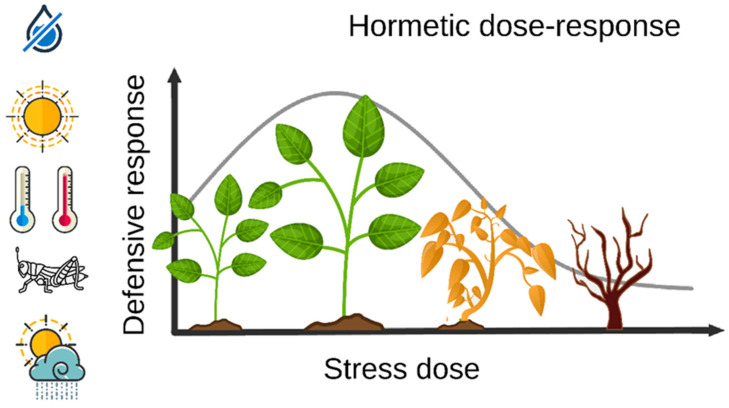
The hormetic behavior of plant stress responses. At low doses, an overcompensation of the damage caused by the stressor increases plant fitness, whereas, at high doses, the stressors disrupt the homeostasis of the organism.

**Figure 2 plants-11-00970-f002:**
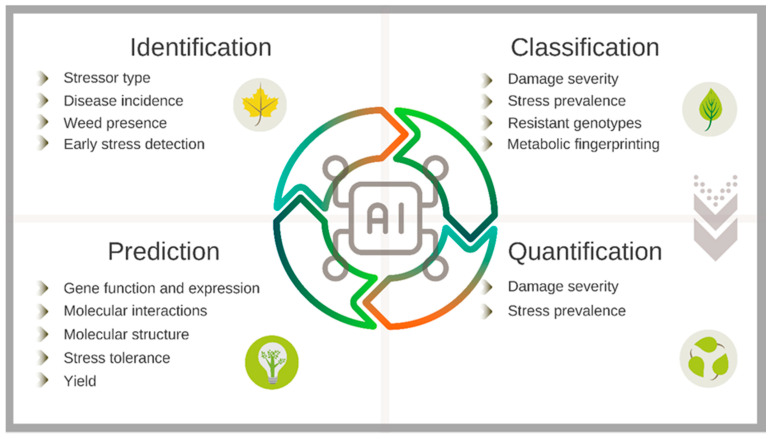
The ICQP paradigm of the four categories for analyzing the stress process of plants. The uses of ML and DL in plant science are summarized in these four general applications. A wide range of datasets can be used for the design of the intelligent algorithms.

**Figure 3 plants-11-00970-f003:**
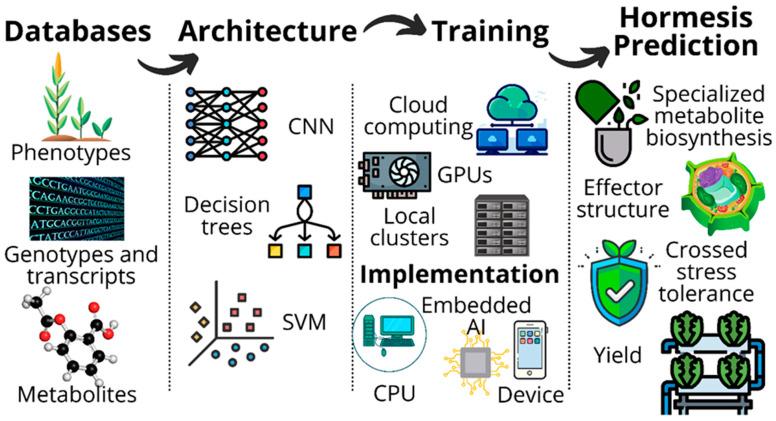
Hormesis characterization through Deep Learning. Plant science uses highly sensitive techniques for detecting variations in gene expression, phenotype, and metabolism caused by environmental interactions. Deep learning, particularly through the implementation of Convolutional Neural Networks (CNN), decision trees, and Support Vector Machine (SVM) algorithms, allows big data processing and interpretation for modeling non-linear biological processes, such as hormesis.

**Figure 4 plants-11-00970-f004:**
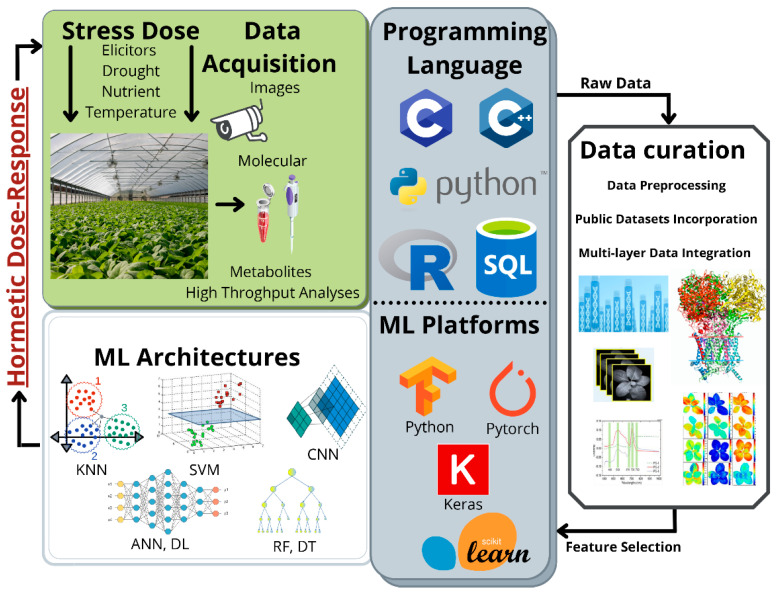
Process of ML implementation for improving hormesis management. Analyzing plant stress responses generates many data, and ML integrates data to model complex systems. Considering the hormetic behavior of plant responses, ML could be used to model dose-response and predict eustress doses, simplifying controlled elicitation in agriculture.

**Table 1 plants-11-00970-t001:** Machine learning-based studies in plant stress under the Identification, Classification, Quantification, and Prediction (ICQP) paradigm.

Artificial Intelligence Technique	Algorithms	(ICQP) Application	Datasets	Model Plant Reported	Stressor	Reference
Deep Learning (image)	Convolutional neural networks (CNN), AlexNet, GoogLeNet, and Inception V3	Identification	1200 images acquired by camera under stress and non-stress conditions	Maize (*Zea mays*), okra (*Abelmoschus esculentus*), and soybean (*Glycine max*)	Water stress	Chandel et al. (2020) [113]
Unsupervised Machine learning	Least squares discriminant analysis (PLS-DA) and least-squares support vector machine (LS-SVM)	Identification	Hyperspectral images of the canopy of tobacco plants	Tobacco	Heavy metal stress Hg	Yu et al. (2021) [114]
Deep Learning (image)	CNN	Identification	1426 images of rice diseases and pests from paddy fields	Rice	Biotic stress	Rahman et al. (2020) [115]
Unsupervised Machine learning (video imaging)	Hidden Markov models (HMMs)	Identification and classification	Chlorophyll fluorescence (ChlF) digital profiles from GrowTech Inc.	*Phaseolus vulgaris* L. (Snap bean)	Stressor “level” groups (low, medium, and high stressed) and three stressor “type” categories (drought, nutrient, and chemical stress)	Blumenthal et al. (2020) [116]
Deep Learning (image)	CNN	Identification and Quantification	1747 smartphones images of arabica coffee leaves.	Arabica coffee	Biotic stress; leaf miner, rust, brown leaf spot, and *Cercospora* leaf spot	Esgario et al. (2020) [117]
Supervised Machine Learning, Partial Least Square Regression, Principal Component Analysis, and combined models	K-nearest neighbors (KNN)	Identification and classification	Spectral signature of leaf samples obtained with a visible, near-infrared spectrometer	Rice	Salt stress	Das et al. (2020) [118]
Supervised Machine Learning	ReliefF, support vector machine (SVM), recursive feature elimination (RFE), and random forest (RF)	Identification and classification	Hyperspectral images from four wheat lines	Wheat	Salt stress	Moghimi et al. (2018) [119]
Deep Learning (image)	CNN	Identification and classification	1575 images (smartphones, compact cameras, DSLR	Different plant specimens	Biotic stress	Arnal Barbedo (2019) [120]
Deep Learning	RF, SVM, multilayer perceptron (MLP)	Identification and classification	Hyperspectral images	*Bromus inermis*	Drought stress	Dao et al. (2021) [121]
Supervised Machine Learning	SVM	Identification and classification	RGB leave images from the Kaggle database	Brinjal leaves	Biotic stress	Karthickmanoj et al. (2021) [122]
Deep Learning (image)	Deep convolutional neural network (DCNN)	Identification, classification, and quantification	Collection of images of stressed and healthy soybean leaflets in the field	Soybean [*Glycine max* (L.) Merr.]	Bacterial blight (*Pseudomonas savastanoi* pv. glycinea), bacterial pustule (*Xanthomonas axonopodis* pv. glycines), sudden death syndrome (*Fusarium virguliforme*), septoria brown spot (*Septoria glycines*), frogeye leaf spot (*Cercospora sojina*), iron deficiency chlorosis, potassium deficiency, and herbicide injury	Ghosal et al. (2018) [123]
Supervised Machine Learning	RF, SVM, KNN	Classification and prediction	Real time terahertz time-domain spectroscopic data (THz-TDS)	Basil, coriander, parsley, baby-leaf, coffee, pea-	Water Stress	Zahid et al. (2022) [124]
Supervised Machine Learning	RF, artificial neural networks (ANN), and	Classification	Multispectral images	Maize	Water stress	Niu et al. (2021) [125]
Supervised Machine Learning	Confident multiple-choice learning	Identification and prediction	Gene expression time-series datasets	*Arabidopsis thaliana*	Heat, cold, salt, and drought	Kang et al. (2018) [126]
Deep Learning (image)	CNN	Classification	Images of Sorghum plant shoot from the Donald Danforth Plant Science Center.	Sorghum plants	Nitrogen deficiency	Azimi et al. (2021) [127]
Supervised Machine Learning	Decision tree (DT), SVM, and Naïve Bayes (NB)	Classification	Metabolite and protein content	*Arabidopsis thaliana*	Metabolic stress	Fürtauer et al. (2018) [128]
Supervised Machine Learning	SVM	Classification	Biweekly RGB, stereo and hyperspectral spatio-temporal images	Sugar beet plants	Abiotic stress conditions (drought and nitrogen deficiency) and one biotic stressor (weed)	Khanna et al. (2019) [129]
Supervised Machine Learning	Hierarchical models	Classification	5916 RGB images (493 plots including Plant Introduction (PI) accessions in different time points)	Soybean (*Glycine max* (L.) Merr.)	Iron deficiency chlorosis	Naik et al. (2017) [130]
Supervised Machine Learning	ANN, CNN, optimum-path forest, KNN, and SVM	Classification	Electrical signal under cold, low light and osmotic stimuli.	Soybean plants	Cold, low light, and osmotic stimuli.	Pereira et al. (2018) [131]
Supervised Machine Learning	RF	Classification	Hyperspectral dataset acquired from the Indian Agricultural Research Institute (IARI)	Wheat	Water stress	Mondal et al. (2019) [132]
Deep Learning (image)	CNN, SVM	Classification	65,184 labeled images from Github resources	Soybean	Biotic (fungal and bacterial diseases) and abiotic (nutrient deficiency and chemical injury) stresses	Venal et al. (2019) [133]
Supervised Machine Learning	MLP and probabilistic neural network (PNN)	Classification	16 maize and 17 wheat genomic and phenotypic datasets with different trait-environment combinations	Maize and Wheat	Drought	González-Camacho et al. (2016) [134]
Supervised Machine Learning	Decision tree (DT), SVM, and NB	Prediction	miRNA concentration.	*Arabidopsis thaliana* plants	Drought, salinity, cold, and heat	Vakilian (2020) [135]
Supervised Machine Learning	Ridge regression, LASSO, elastic net, RF, reproducing kernel Hilbert space, Bayes A and Bayes B	Prediction	A set of 29,619 cured Single Nucleotide Polymorphisms, genotyped across a panel of 240 maize inbred lines	Maize	Drought stress	Shikha et al. (2017) [136]
Deep Learning	CNN	Prediction	Three maize and six wheat data sets.	Maize and wheat	Environmental stress	Montesinos-López et al. (2018) [137]
Supervised Machine Learning	Genomic random regression	Prediction	Complete genotypes, molecular markers, and phenotypic traits of stressed and control groups.	Wheat	Environmental stress	Ly et al. (2018) [138]

**Table 2 plants-11-00970-t002:** Deep Learning architecture, hardware, and applications.

DL Architecture	Application	Hardware	Reference
Deep Neural Networks	Toxicity Prediction	Nvidia Tesla K40	Mayr et al. (2016) [150]
Convolutional Neural Network	Photosynthetic pigments Prediction	CPU core i5 1.6 GHz, 8 GB DDR3 RAM, GPU not specified	Prilianti et al. (2020) [151]
Convolutional Neural Network	Pigments Prediction	Nvidia GTX 1020Ti, Intel Xeon W-2133, 32 GB	Mu et al. (2020) [152]
AlexNet and SqueezeNet	Plant Disease Detection	Nvidia Jetson TX1	Durmus et al. (2017) [153]
Convolutional Neural Network	Plant Disease Detection	Nvidia GTX1080	Ferentinos (2018) [154]
Convolutional Neural Network	Plant Disease Detection	Nvidia Tesla K40c	Too et al. (2019) [155]
Deconvolutional Neural Network	Plant Disease Detection	Nvidia GeForce Titan X, Intel Core I7 3.5 GHz	Wang et al. (2017) [156]
Point Completion Network	Plant Phenotyping	Nvidia Titan V. Xeon Gold 6146 3.20 GHz, 128 GB RAM	Wu et al. (2019) [157]
Deep Convolutional Neural Network	Predicting Phenotypes from Genotypes	Nvidia GeForce TITAN-XGPU	Ma et al. (2018) [158]
U-net	Phenotyping and Plant Growth	Nvidia Tela V100	Tausen et al. (2020) [159]

## Data Availability

Not applicable.

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
