# Peer review of "Machine Learning for Plant Stress Modeling: A Perspective towards Hormesis Management"

_plants, 2022, doi:10.3390/plants11070970_

Round 1
Reviewer 1 Report
The effort made by the authors is very valuable, as nowadays it is impossible to follow most of the current literature on a topic of interest. This paper is a comprehensive review focused on the Machine Learning and Deep Learning applications in plant stress science for improving the development of hormesis management protocols. The manuscript fits within the scope of the journal. This review deals with a very interesting topic and it is structured and written in an appealing way. However, here are some suggestions for future improvement:
Please include some information about the work method: How do you search literature data? How was the period? Which sources? English language and style are fine but minor spell check is required. Check the bibliography to be written according to the requirements of the journal (See for ex L331). There are some paragraphs without including the citation (For ex: L153-154; 178-179, 268-270.
Author Response
Dear Reviewer 1
Thank you very much for your suggestions, in the attached file you will find the answers.
Best regards

Reviewer 2 Report
Comments on plants-1648189
The authors have chosen a very interesting topic for review. The following additions might improve the understanding and quality of the subject matter under consideration
There are various typos and grammatical mistakes, which need a thorough readout of the manuscript, especially from a native English speaker
Regarding the application of these machine learning approaches, a simple flowchart diagram may be added that how can we apply these approaches under real conditions by giving some examples with some explanation. In addition, discussion about the software that is helpful may also be added
During the discussion of different modeling approaches, the role of simulation through the Mamdani Fuzzy Logic technique may also be explained in a paragraph. The following papers may be consulted and cited
https://doi.org/10.1016/j.chemosphere.2019.04.022
and for ANN
https://doi.org/10.1007/s00477-015-1125-2
Table 1 Please confirm “IA Technique”
The authors are highly suggested to add the limitations of different machine learning approaches under certain conditions with suggestions of some remedies for these limitations
Future recommendation or future outlook needs to be added
Author Response
Dear Reviewer 2
Thank you very much for your suggestions, in the attached file you will find the answers.
Best regards

Round 2
Reviewer 2 Report
The authors have incorporated all the suggestions/comments. The paper may be accepted